# Clinical and Therapeutic Intervention of Hypereosinophilia in the Era of Molecular Diagnosis

**DOI:** 10.3390/cancers16071383

**Published:** 2024-03-31

**Authors:** Lynh Nguyen, Aditi Saha, Andrew Kuykendall, Ling Zhang

**Affiliations:** 1Department of Pathology, James A. Haley Veterans’ Hospital, Tampa, FL 33612, USA; 2Department of Malignant Hematology, H. Lee Moffitt Cancer Center, Tampa, FL 33612, USAandrew.kuykendall@moffitt.org (A.K.); 3Department of Pathology, H. Lee Moffitt Cancer Center, Tampa, FL 33612, USA

**Keywords:** hypereosinophilia, hypereosinophilic syndrome, myeloid/lymphoid neoplasms, tyrosine-kinase gene rearrangement, molecular diagnosis, targeted therapy

## Abstract

**Simple Summary:**

Hypereosinophilia (HE) is defined as an elevated peripheral eosinophilic count >1.5 × 10^9^/L. It constitutes a broad spectrum of secondary non-hematologic disorders and primary hematologic processes with heterogenous clinical presentations, a number of subclassifications (familial, idiopathic, hypereosinophilic syndrome [HES], myeloid/lymphoid neoplasms, organ restricted, or with unknown significance) and some can have potential lethal outcome from end-organ damage, necessitating timely and accurate diagnosis. Treatment guidelines are established for patients with HE based on its clinical presentation and risk stratification. Observation is recommended for patients who have mild hypereosinophilia and lack organ damage-related signs and symptoms while tyrosine kinase inhibitors are offered to patients harboring *PDGFRA*, *PDGFRB*, *FGFR1*, *JAK2* or *FLT3* rearrangements. Additionally, corticosteroids are considered as the mainstay of therapy, hydroxyurea, and cytokine blockage (e.g., mepolizumab) have been used for lymphocytic-variant HE, a second line therapy for steroid-resistant cases of HE, and as a novel targeted therapy for HES.

**Abstract:**

Hypereosinophilia (HE) presents with an elevated peripheral eosinophilic count of >1.5 × 10^9^/L and is composed of a broad spectrum of secondary non-hematologic disorders and a minority of primary hematologic processes with heterogenous clinical presentations, ranging from mild symptoms to potentially lethal outcome secondary to end-organ damage. Following the introduction of advanced molecular diagnostics (genomic studies, RNA sequencing, and targeted gene mutation profile, etc.) in the last 1–2 decades, there have been deep insights into the etiology and molecular mechanisms involved in the development of HE. The classification of HE has been updated and refined following to the discovery of clinically novel markers and targets in the 2022 WHO classification and ICOG-EO 2021 Working Conference on Eosinophil Disorder and Syndromes. However, the diagnosis and management of HE is challenging given its heterogeneity and variable clinical outcome. It is critical to have a diagnostic algorithm for accurate subclassification of HE and hypereosinophilic syndrome (HES) (e.g., reactive, familial, idiopathic, myeloid/lymphoid neoplasm, organ restricted, or with unknown significance) and to follow established treatment guidelines for patients based on its clinical findings and risk stratification.

## 1. Introduction

Eosinophils are end-stage granulocytes derived from myeloid progenitors in the bone marrow in response to stimulation by cytokines, (e.g., interleukin [IL]-3, IL-5, and granulocyte-macrophage colony-stimulating factor [GM-CSF]). After terminal differentiation, mature eosinophils are released into peripheral blood and either stay in circulation or infiltrate surrounding tissue. Mature eosinophils have a short half-life (5–8 h, ranging from 3 to 24 h) in the blood, but can be retained in tissue for several days (half-life of 8–12 days), ranging from 1.5 days in lungs to 6 days in small intestines, thymus and uterus [1]. Morphologically, eosinophils show coarse, round, orangish granules, distinguishing them from neutrophils. Eosinophils play a significant role in the innate and adaptive immune systems by releasing various cytokines and cytoplasmic granules or by direct cell interaction. They modulate the inflammatory response involving tissue remodeling and repair, defense against parasites, and allergic reactions [2].

Normally, eosinophils only account for less than 5–6% of the circulating white blood cells and usually absolute counts are <0.5 × 10^9^/L in the blood. The level of eosinophils is strictly regulated by the body’s cytokines. A circulating eosinophil count of ≥0.5 × 10^9^/L is defined as eosinophilia. An increase of eosinophils with an absolute eosinophil count greater than 1.5 × 10^9^/L is defined as hypereosinophilia (HE) [3].

Hypereosinophilic syndrome (HES) is a constellation of clinical presentations with sustained HE on two occasions at least one month apart or marked tissue eosinophilia associated with tissue damage or dysfunction. Symptoms may be mild presenting as weakness or fatigue or be life threatening as a result of end organ damage or dysfunction as seen with endomyocardial fibrosis and thromboembolism [4]. Caution should be taken to excluded other etiologies that may result in organ damage.

Tissue HE can be visualized with light microscopy on biopsy. However, the thresholds for eosinophilia are not well established in all tissue types. Having one or more of the following features may meet criteria for tissue HE: (A) for example, in diagnosing eosinophilic esophagitis, the minimal requirement is >15 eosinophils per high-power field associated with basal layer hyperplasia [5]; (B) within the bone marrow, 20% of nucleated cells must be eosinophils [6]; (C) criteria can also be met if there is extensive eosinophilic infiltration or marked deposition of eosinophilic granules or eosinophilic degranulation as detected by immunofluorescence or monoclonal antibody for eosinophil peroxidase in the opinion of the reviewing pathologist [6]. 

HE is composed of a wide spectrum of clinical scenarios, including primary (clonal proliferation, hematologic) or secondary (reactive, non-hematologic) processes (Table 1). Reactive hypereosinophilia is more common than clonal or primary HE. Therefore, it is important to obtain a thorough clinical history and exclude secondary or reactive causes of eosinophilia before diagnostic evaluation for primary HE. This review will focus on the differential diagnoses of common variants of HE and discuss potential therapy options for primary HE based on advanced molecular approaches. 

## 2. Epidemiology

Given the various causes of HE, its incidence, prevalence, and subclassification are not well documented. If defining eosinophilia as an absolute eosinophil count ≥0.45 × 10^9^/L, the prevalence is approximately 1–2% in the general population [7]. In the United States, HE is uncommon with an incidence estimated at 0.3–6.3 per 100,000 individuals [8]. According to the data collected in the Surveillance, Epidemiology, and End Results (SEER), the estimated incidence of HES including chronic eosinophilic leukemia (CEL) in age-adjusted population is <1 per 100,000 [9]. The demographics for individuals affected it is quite different for specific variants of HE. For example, HES affects mostly young to middle-aged patients (20–50 years of age); however, variants associated with immunodeficiency can be seen in children. These children are likely to present with higher eosinophil counts and gastrointestinal (GI) symptoms [10]. Other HE variants, like myeloid/lymphoid neoplasms (MLN) with *PDGFRA*/*PDGFRB* rearrangements are almost exclusively seen in males while other variants like T-lymphocytic variants of HE (L-HES) show no gender preference [11,12]. 

## 3. Etiology

The etiology of HE can be categorized as primary, secondary, and idiopathic. Primary etiologies occur when clonal eosinophilic expansion results from specific genetic mutations or rearrangements in hematopoietic cells. In secondary or reactive HE, the expansion of eosinophils is a direct result of cytokine overproduction (e.g., IL-5) that can often occur in parasitic infections, solid tumors, T-cell lymphomas, and some connective tissue diseases. Any medication or supplement can cause a drug hypersensitivity reaction or drug-associated eosinophilia. Although individuals affected can be asymptomatic, some can present systemically with interstitial nephritis, eosinophilic hepatitis, or even eosinophilia-myalgia syndrome [13]. Secondary HE can also be associated with specific clinical syndromes like Churg-Strauss (eosinophilic granulomatosis and polyangiitis) and immunodeficiencies. These eosinophils, in contrast to those in primary HE, are polyclonal. Lastly, idiopathic HES (iHES) occurs when known causes of primary and secondary HE have been excluded and the underlying cause is unknown. Some patients can present with HE without explanation or with clinical complications related to tissue damage. These cases are subcategorized as HE of unknown significance (HE_US_).

## 4. Pathogenesis

In general, the receptors of eosinophils play a critical role in the body’s functional response to antigens or clonal proliferation. (1) IL-5 receptor alpha subunit: IL-5 is produced by eosinophils, basophils, NK-cells, and mast cells via activation of T-helper (Th_2_) cells. Together with IL-2, IL-13 and chemokines ligands (CCL11, CCL24, CCL26), IL-5 and the IL-5 receptor promote maturation of eosinophilic precursors, mediate activation of eosinophils, prolong survival of eosinophils in circulation, and recruit mature eosinophils to the tissue, leading to an inflammatory infiltrate. (2) Chemokine receptor-3 (CCR3): CCR1 is the receptor for CCL3 and CCL5 and are expressed on eosinophils and considered the platelet-activating factor receptor. Because of activation by CCL5 and other cytokines, CCL3 can stimulate eosinophilic chemotaxis, which is important for eosinophil migration. (3) Lesser known are sialic acid-binding immunoglobulin (IG)-like lectin (SIGLEC-8) and pattern recognition receptors expressed by human eosinophils. SIGLEC-8 is a cell-surface IG-like lectin which may affect elective apoptosis of eosinophils, while several protein families (e.g., toll-like receptors [TLR]) constitute pattern recognition receptors on eosinophils. One study showed binding TLR7 ligand R837 on eosinophils increased IL-8 production. Additionally, triggering TLR7 release is one of the key steps in adhesion, migration, and prolonged survival of eosinophils. 

In addition to responding to signals, eosinophils also interact with other granulocytes, T-cells, B-cells, dendritic cells via MHC class II and costimulatory molecules, recruiting Th_2_ cells, releasing CCL17, CCL22, a proliferation-inducing ligand (APRIL) and IL-6, etc. Eosinophils also play a key role in maturing and activating dendritic cells. Additionally, they can produce major basic proteins for activation of neutrophils and mast cells; IL-4 and IL-13 for activation of macrophages, eosinophil cationic protein; eosinophil peroxidase for activation of mast cells; and eosinophil-derived nerve growth factor for prolonging mast cell survival. (Figure 1) [14]. 

Taken together, overproduction of eosinophils may be caused by a molecular defect leading to altered signaling, or overproduction of IL-5 or other cytokines that could stimulate eosinophil production and survival. Given the complexity of functions and the interaction of eosinophils with other inflammatory cells as well as the degree of cytokine release via different pathways, the clinical course of HE is very heterogeneous. Eosinophils infiltrate and in cases of HES can damage the skin, lungs, GI tract and less commonly the heart and brain. Defects in suppression of eosinophils, prolonged survival of eosinophils, and enhanced eosinophilic activity could be other potential explanations for HE although have not been thoroughly investigated.

The mechanisms of tissue damage are a consequence of eosinophil infiltration, secondary tissue fibrosis, allergic reaction, ischemia, and hypercoagulability or thrombosis induced by the release of eosinophilic granules [15].

## 5. Diagnosis

### 5.1. Clinical

The clinical manifestation of HE is heterogenous, varying from asymptomatic to systemic or syndromic secondary to organ damage. The diagnostic approach should follow the guidelines proposed by the fifth edition of the World Health Organization (WHO) Classification of Haematolymphoid Tumours and updated International Eosinophilia Society (IES) (Figure 2) [3]. Secondary eosinophilia should always be investigated first. The clinical history must be carefully reviewed for allergies, current medications, and infections, particularly for the patients with immunocompromised status, (e.g., patients with HIV with tissue-invasive strongyloidiasis). After excluding secondary causes of eosinophilia, a systemic investigation for primary HE should be conducted [3]. Imaging studies (positron emission tomography [PET] or computed tomography [CT] scan) are recommended to evaluate for L-HES, suspected organomegaly, and hematolymphoid or non-hematologic neoplasms. Serum troponin T or I, echocardiogram, imaging studies and pulmonary function tests are commonly ordered to evaluate organ dysfunction associated with HE, in particular HES. Genetic counseling should be considered for patients with suspected familial HE.

### 5.2. Laboratory and Pathology

Complete blood count (CBC) with differential counts, including absolute eosinophil count and duration of elevation should be closely monitored.

To exclude secondary HE, laboratory tests for infectious causes including cultures for microorganisms, serology, or polymerase chain reaction (PCR) for viruses and fungi, and identification of parasites should be part of the initial evaluation. Serological testing for strongyloidiasis is usually requested in immunocompromised patients. An immunology profile should be ordered for individuals with inborn errors of immunity (IEIs) or known family history of hereditary HE. Immunoglobulin levels would be useful in those with IgG4-related disorders (elevated IgG4), suspected parasitic infection, hypersensitivity disorder or hyper IgE syndrome (elevated immunoglobulin E [IgE]). Additionally, a comprehensive metabolic chemistry panel and liver function tests are helpful in assessing organ dysfunction. 

To confirm diagnosis of primary HE and begin potential target therapy, a more comprehensive, systemic approaching is necessary which should include: (1) morphologic evaluation of peripheral blood, bone marrow, and immunophenotyping by flow cytometry analysis or immunohistochemistry; (2) conventional cytogenetics; (3) fluorescence in situ hybridization (FISH); (5) molecular studies with PCR; and (6) next generation sequencing (targeted gene mutation panel or RNA sequencing for fusion products) to identify histopathologic and clonal evidence of an acute or chronic hematolymphoid neoplasm (see Clinical Variants section). Serum tryptase levels should be drawn if clinically suspicious for systemic mastocytosis (SM) with or without associated myeloid neoplasms. T-cell and/or B-cell gene rearrangements detected by PCR assess lymphocyte clonality and flow cytometry analysis aids in detecting abnormal T-cell subsets (CD3^−^/CD4+ or CD3^+^/CD4^−^/CD8^−^) like those seen in L-HES [11]. 

One should be careful making a diagnosis solely based on the morphology of eosinophils in peripheral blood or bone marrow as it may be misleading. Cytologically abnormal eosinophils with increased size, cytoplasmic vacuoles, or abnormal granulation or nuclear lobation can be seen in both reactive and neoplastic conditions. Eosinophil morphology may not be reliable; however, examination of the peripheral blood, bone marrow aspirate smears, and core biopsy together may help narrow the diagnosis, especially with elevated blast counts, presence of parasitic infection, or evidence of lymphoma. Bone marrow biopsy is essential in diagnosing a myeloproliferative variant of HE (M-HES), while tissue biopsy is indicated for suspected organ damage or organ restricted HE. 

Lastly, a diagnosis of iHES should only be made after a thorough investigation of known primary and secondary causes. Assessing tissue damage and organ dysfunction can be performed with clinical signs and symptoms, tissue biopsy, and specific tests (see Table 2). The diagnostic algorithm is shown in Figure 2.

### 5.3. Potential Genetic Determinants and Biomarkers

Extracellular vesicles and a number of potential blood-based biomarkers are being investigated to aid in diagnosis and treatment of atopic conditions [16]. Eosinophils have been noted to carry microRNA (miRNA) in extracellular vesicles to other cells, potentially playing a role in gene regulation and expression [16]. One study sequenced the eosinophil transciptome in individuals with atopic conditions and compared them to healthy controls. They discovered at least 18 miRNAs were differentially expressed in individuals with allergic conditions when compared to those unaffected [17]. Additionally, genome-wide association studies (GWAS) have identified multiple genes (chromosomes 1q23 [FCER1A], 5q31 [RAD50, IL13, IL4], 12q13 [STAT6], loci 6p21.3 [HLA-DRB1] and 16p12 [IL4R, IL21R]) that may influence the regulation of IgE and serum IgE levels [18]. Interestingly, the relationships of total serum IgE levels and atopic conditions may not be as simple as initially thought. Specific atopic conditions (asthma, allergies rhinitis, atopic dermatitis) show little overlap with genetic determinants of total serum IgE levels, suggesting that elevated serum IgE levels may be an epiphenomenon [19,20,21]. A stronger association may be seen in specific IgE with allergic conditions rather than total serum IgE levels [22,23,24]. Studies are ongoing, but potential genetic markers and biomarkers are on the horizon.

## 6. Differential Diagnosis

The differential diagnosis for eosinophilia is vast, encompassing reactive and neoplastic conditions. A good clinical history is paramount for determining the etiology. Being familiar with clinical variants of HE is critical in narrowing down the differential diagnoses of HE. The clinical features, in addition to laboratory, pathology and genetic findings are briefly illustrated in Table 3. 

### 6.1. Myeloproliferative Variants of Hypereosinophilia (M-HES)

Myeloid or lymphoid neoplasms with eosinophilia can demonstrate gene rearrangements in *PDGFRA*, *PDGFRB*, *FGFR1*, *JAK2*, or *FLT3* or *ETV6::ABL* or other tyrosine kinase gene fusion (MLN-TK). These are collectively recognized in the fifth edition of the WHO classification of Hematolymphoid Tumours as myeloid/lymphoid neoplasms (MLN) with eosinophilia and defining gene rearrangements. 

There is a large spectrum of clinical manifestations for MLN-TK, which may mimic CEL, other myeloproliferative neoplasms (MPN), systemic mastocytosis (SM), myelodysplastic neoplasm (MDS), myelodysplastic/myeloproliferative neoplasms (MDS/MPN), T- or B-lymphoblastic leukemia, or acute myeloid leukemia (AML). Before TK-related gene rearrangements were identified, MLN-TK were likely misdiagnosed especially with such a heterogeneous clinical and morphologic presentation on tissue or bone marrow biopsy where eosinophilia may be absent. Concurrent or subsequent development of MLN should trigger an investigation of MLN-TK as the cause. Routine karyotyping may or may not be helpful given the various TK genes and their numerous gene partners and cryptic alterations. FISH studies and RNA sequencing have become essential in detection, utilizing specific gene probes for specific gene translocations. 

MLN-TK originate from pluripotent hematopoietic stem cells that give rise to neutrophils, eosinophils, monocytes, mast cells, and lymphocytes. Each key TK gene harbors multiple partner genes (Table 4). *FIP1L1* is the most common partner for *PDGFRA*. *FIP1L1::PDGFRA* leading to a cryptic deletion of chromosome 4q12, which is the most common molecular aberration in this group and can be detected by FISH, but not by conventional karyotyping. Patients present with elevated serum vitamin B_12_ and tryptase levels along with prominent peripheral blood and/or tissue eosinophilia. In general, bone marrow is usually hypercellular with an increase in eosinophils and mast cells and may show reticulin fibrosis in cases without leukemic transformation. Overall prognosis for these patients is favorable with imatinib treatment. 

MLN with *PDGFRB* rearrangement involves chromosomal rearrangements of 5q32 leading to formation of a *PDGFRB* fusion gene. Its most common partner gene is *ETV6*, but other partner genes and variants have been described. Like MLN with *PDGFRA* rearrangement, it is extremely sensitive to TK inhibitors (TKI). Marked eosinophilia with accompanying neutrophilia and monocytosis is common. Overall survival is approximately 90% over 10 years; however, patients presenting with complex cytogenetics have worse outcomes [27,28]. 

Similar to other MLN-TK, *FGFR1* rearrangements can present with varying phenotypes and clinically manifested as B-cell or T-cell lymphoblastic leukemia/lymphoma, AML, mixed phenotype leukemia, MPN, or MDS/MPN due in part to the different *FGFR1* fusion partner genes and their effect on the intracellular signaling pathways. For example, *ZMYM2::FGFR1* most commonly presents as a T-lymphoblastic leukemia/lymphoma, *BCR::FGFR1* and *TPR::FGFR1* present histologically similar to chronic myeloid leukemia (CML), and *CEP43::FGFR1* and *CNTRL::FGFR1* show features similar to chronic myelomonocytic leukemia (CMML). Prominent eosinophilia with or without neutrophilia or monocytosis is noted in the peripheral blood. Unlike *PDGFRA* and *PDGFRB*, patients with MLN with *FGFR1* rearrangements have an aggressive clinical course with blast transformation within 1–2 years of diagnosis [3]. 

MLN with *JAK2* rearrangement is an emerging entity. *PCM1* is its most common fusion partner. Other partner genes described include but are not limited to *BCR* and *ETV6* [29]. Clinically patients present as a MPN or MDS/MPN with neutrophilia and/or monocytosis with varying degrees of eosinophilia and rarely with HE. Those with *ETV6::JAK2* commonly present as B-ALL though *BCR::JAK2* presents with MDS with neutrophilia or B-ALL [29,30]. In patients with *PCM1:JAK2*, the bone marrow is typically hypercellular demonstrating eosinophilia, erythroid hyperplasia with dyserythropoiesis, and myelofibrosis [30]. Prognosis is variable with some patients presenting with indolent disease and others presenting with a more aggressive clinical course [3].

Patients with MLN with *FLT3* rearrangement often present with leukocytosis with or without eosinophilia and monocytosis and frequently show extramedullary involvement. *ETV6* is the most common fusion partner [31]. Histologically, the marrow may present with features of MDS, MPN, CMML with blasts of any of cell origin (myeloid, B-cell, or T-cell). These patients usually have an aggressive clinical course [32]. 

MLN with *ETV6::ABL1* may show overlapping histologic features with CML, but can also present as an MDS/MPN with neutrophilia or a CEL. Prognosis is poor for these patients [3].

*KIT*-mutated SM can also present with clonal eosinophilia. Patients with *KIT* p.D816V mutation are not sensitive to imatinib therapy, but may respond to midostaurin, a multitargeted protein kinase inhibitor or avapritinib, a potent *KIT* D816V inhibitor [33]. Bone marrow shows mast cell infiltrates consisting of aggregates of ≥15 mast cells in which >25% are atypical or spindle-shaped and show aberrant expression of CD2, CD25 and/or CD30. Serum tryptase levels are persistently >20 ng/mL. Mutation burden predicts patient survival and risk of progression. Eosinophils may be increased in the marrow, which may be secondary reaction or part of the neoplastic clone [3] (Table 4).

**Table 4 cancers-16-01383-t004:** Clinical and molecular features of MLN-TK.

	Partners	Concurrent Mutations	Typical Clinical Association	Reference (PMID)
*PDGFRA* *	*FIP1L1*, *KIF5B*, *CDK5RAP2*, *STRN*, *ETV6, BCR*, *TNKS2*	N/A	MPN or MDS/MPN typically in chronic phase and less frequently in blast phase of myeloid or lymphoid lineage	[34,35,36,37,38,39,40,41]
*PDGFRB*	*WDR48*, *CAPR1N1*, *TPM3*, *PDE4DIP*, *SPTBN*, *PRKG2*, *GOLGA4*, *TNIP1*, *HIP1*, *HECW1*, *KANK1*, *CCDC6*, *SART3*, *GIT2*, *ERC1*, *BIN2*, *NIN*, *CCDC88C*, *TP53BP1*, *NDE1*, *RABEP1*, *SPECC1*, *MYO18A*, *COL1A1*, *DTD1*	N/A	Commonly CMML with eosinophilia and less commonly MDS/MPN with neutrophilia (formerly aCML), and CEL (or MPN with eosinophilia)	[42,43,44]
*FGFR1*	*ZMYM2*, *FGFR1OP*, *TRIM24*, *MYO18A*, *HERVK*, *FGFR1OP2*, *RANBP2*, *LRRFIP1*, *CUX1*, *CPSF6*, *BCR*, *TPR*, *CEP43, CNTRL*	Concurrent mutations involving *RUNX1;* associated with increased proliferation of the clone and poor outcome	Variable phenotype including precursor B-cell, T-cell, myeloid or MPAL or MPN or MDS/MPN with associated eosinophilia, rarely B-ALL	[45,46,47,48,49,50,51,52]
*JAK2*	*PCM1*, *ETV6*, *BCR*	*ASXL1*, *BCOR*, *ETV6*, *RUNX1*, *SRSF2*, *TET2*, and *TP53*	MPN, ALL, AML	[53]
*FLT3*	*BCR*, *ZMYM2*, *TRIP11*, *SPTBN1*, *GOLGB1*, *CCDC88C*, *ZBTB44*, *MYO18A*	*ASXL1*, *SETBP1*, *U2AF1*, *STAT5B*, *TP53*, *SRSF2*, *TET2*, *RUNX1*, and *PTPN11*	Extramedullary involvement with T-ALL, MyeS, or rarely with mixed-phenotype features including B-cell, T-cell, or myeloid lineage disease.	[31,54,55]
*ETV6::ABL1*	*ETV6::ABL1*	N/A	MDS/MPN with neutrophilia, CEL, or other MDS/MPN.	[56,57,58,59]
Other	*ETV6::FGFR2; ETV6::LYN; ETV6::NTRK3; RANBP2::ALK; BCR::RET* and *FGFR1OP::RET*	N/A	MDS/MPN, often with notable eosinophilia, ±monocytosis, T-cell differentiation is more common such as T-ALL or PTCL, mast cell proliferations and/or bone marrow fibrosis	[60,61,62,63,64,65]

Abbreviations: aCML, atypical chronic myeloid leukemia; ALL, acute lymphoblastic leukemia; AML, acute myeloid leukemia; B-ALL, B-cell acute lymphoblastic leukemia; CEL, chronic eosinophilic leukemia; CMML, chronic myelomonocytic leukemia; MDS, myelodysplastic syndromes; MDS/MPN, myelodysplastic/myeloproliferative neoplasms; MLN, myeloid/lymphoid neoplasms; MPAL, mixed-phenotype acute leukemia; MPN, myeloproliferative neoplasms; Mye-S, myeloid sarcoma; N/A, not applicable; PTCL, peripheral T-cell lymphoma; T-ALL, T-cell acute lymphoblastic leukemia; TKs, tyrosine kinases. * *PDGFA* active domain mutation can also lead to the MLN according to fifth edition. WHO Classification of Haematolymphoid Tumours. *JAK-STAT* signaling pathway is the key mode of signal transduction. Activation of *STAT5* is triggered by a vast variety of cytokines and growth factors. A recurrent *STAT5b* N642H mutation is recently identified in a subset of M-HES and plays a key role in pathogenesis. Detection of the unique mutation sheds a light in diagnosis and potential targeted therapy for M-HES [66].

### 6.2. T-Lymphocytic Variants of HE (L-HES)

There is a group of patients who show an expansion of aberrant T-cells and eosinophilia excluding other explainable etiologies [67]. The prevalence of a lymphocytic variant of HES is not well defined. In L-HES, IL-5-producing T-cells have been identified in the peripheral blood [68]. Most commonly, these aberrant T-cells are CD3^−^/CD4^+^, but other phenotypes have been described (e.g., CD3^+^/CD4^+^/CD7^−^ and CD3^+^/CD4^−^/CD8^−^) [69]. Aberrant loss of CD7 or CD27 has been reported [67]. An aberrant CD3-gamma gene transcription may cause the lack of CD3 expression [70]. These patients frequently have skin and soft tissue involvement, lymphadenopathy, and have elevated serum IgE. Skin manifestations are variable, which can present as a rash, macular-papular lesions, or erythematous changes. Superficial enlargement of the lymph nodes is one of the most common manifestations of systemic disease [69], warranting biopsy and immunophenotyping. A third of these patients have elevated lymphocyte counts [69].

Not much is known about the molecular mechanism. T-helper type 2 (Th_2_) cytokines (IL-4, IL-13 and GM-CSF) may be involved in the increased serum IgE production and polyclonal hypergammaglobulinemia [71,72,73,74]. One recent study identified a somatic mutation in *STAT5* [75] though another study described a gain of function mutation in *STAT3* [76,77]. Distinguishing L-HES from T-cell lymphoma is important because L-HES usually follows an indolent course, have mixed reactive and clonal lymphocyte populations, and respond well to corticosteroids [70]. However, transformation to T-cell lymphoma may occur and patients should be followed regularly [78]. Evaluation using only PCR for T-cell clonality is insufficient to differentiate the two entities. If possible, additional laboratory studies should be utilized to confirm elevated levels of Th_2_ cytokines in these patients [3,79].

### 6.3. Idiopathic HES (iHES)

Cases of HE with end organ damage in which the etiology is unknown even after careful evaluation fall under the category of iHES [80]. These patients should have elevated eosinophilic count >1.5 × 10^9^/L for at least 6 months associated with tissue damage. If tissue damage is not identified, it then should be called idiopathic HE (iHE).

Importantly, reactive causes of eosinophilia, myeloid malignancies associated eosinophilia (including AML, MPN, MDS, MDS/MPN and SM), and the presence of an aberrant T-cell population (L-HES) must be excluded. The presence of a clonal genetic or molecular aberration, in conjunction with abnormal bone marrow findings favor a diagnosis of CEL according to the fifth edition of the WHO Classification of Haematolymphoid Tumours. 

### 6.4. Hypereosinophilia of Undetermined Significance (HE_US_)

Patients with persistent HE without HE defined organ damage who do not meet diagnostic criteria for iHES, or familial HES, and have no known reactive or neoplastic disorders that could lead to HE are subclassified as HE of undetermined significance (HE_US_) [6]. Essentially, HE_US_ is a diagnosis of exclusion. Those that fall in this category could potentially evolve into any of the already described specific subtypes of HE. Thus, close clinical monitoring is warranted.

### 6.5. Familial HE/HES

Familial HE/HES includes a list of hereditary disorders and syndromes that often present in childhood usually with accompanying. Familial HES shows an autosomal dominant inheritance. Patients and affected family members typically present with marked eosinophilia with some end organ damage. Eosinophilia can begin as early as four months of age. Some may remain asymptomatic, but others fatally progress to endomyocardial fibrosis. Variants of familial HES can be associated with single organ damage [81,82,83]. The gene responsible for familial HES has been mapped to chromosomal 5q31-q33 [84]. Interestingly, the mutation does not involve genes encoding IL-3, IL-4, IL-5, IL-13 or GM-CSF. Many children with IEIs manifest with life-long mild eosinophilia without developing HES.

In a study conducted by the National Institute of Health (NIH) on familial HES, 13 affected (having >1.5 × 10^9^ eosinophils/L at least twice, 6 months apart) were compared to 11 unaffected family members. Levels of major basic protein and eosinophil derived neurotoxin were elevated in patients with familial HES. Similarly, increased expression of CD25, CD69, and HLA-DR (activation markers) were detected on flow cytometry in the affected when compared to unaffected family members. However, compared to patients with non-familial HES, the levels of eosinophil granule proteins and activation markers were not as high, suggesting those with familial HES may have a benign clinical course due to less eosinophil activation [85].

### 6.6. Specific or Defined Syndromes Associated with HE

HE can be associated with primary immunodeficiency disorders that show autosomal dominant or recessive or X-linked inheritance [6]. The defects can be caused by (1) antibody deficiencies (e.g., common variable immunodeficiency disorder), or combined immunodeficiencies (e.g., Omenn syndrome, Wiskott-Aldrich syndrome, Netherton syndrome, or hyper IgE syndrome); (2) diseases associated with dysregulation of cellular immunity (e.g., autoimmune lymphoproliferative syndrome [ALPS]); or (3) dysregulation of phagocytosis (e.g., severe congenital neutropenia, formerly Kostmann disease) [6]. 

Disorders associated with immune dysregulation can also present with eosinophilia > 1.5 × 10^9^/L. The etiology of eosinophilia is unclear, but conditions include inflammatory bowel disease, sarcoidosis, *CARD9* deficiency, collagen-vascular disease, IgG4-related disease, HIV/AIDS infection, and hyper IgE syndromes like *DOCK8* deficiency [86]. Gleich syndrome, also known as episodic angioedema with eosinophilia, presents clinically with recurrent episodes of angioedema, fever, pruritus, and weight gain with laboratory findings showing elevated serum IgM and marked eosinophilia in the blood [87]. Other diseases associated with eosinophilia include eosinophilic granulomatosis with polyangiitis (Churg-Strauss syndrome), a disease limited to the blood vessels [88] and Loeys-Dietz syndrome, a rare genetic multisystem connective disorder that commonly shows eosinophilic gastrointestinal disease and in severe cases predisposes individuals to aortic aneurysms and dissections [89,90].

### 6.7. Organ-Restricted HE Conditions

Organ-restricted HE conditions demonstrate a peripheral eosinophilia > 1.5 × 10^9^/L and involve a single organ. This disorder includes eosinophilic esophagitis, eosinophilic gastrointestinal disorders, eosinophilic dermatitis, chronic eosinophilic pneumonia, and eosinophilic cellulitis (Wells’ syndrome) [6] and can be difficult to distinguish from iHES.

### 6.8. Secondary/Reactive HE

Reactive conditions must be excluded as the cause of HE. Allergies and atopic conditions, infections (parasitic, fungal, etc.), collagen-vascular disease (Churg-Strauss syndrome, Kimura disease), cyclical eosinophilia, Löffler syndrome, and angiolymphoid hyperplasia can all present with eosinophilia. Administration of cytokines IL-2, IL-3, IL-5, or GM-CSF as medication can also elevate the number of eosinophils in blood. Additionally, any medication can cause a drug hypersensitivity reaction or drug-associated eosinophilia. Eosinophilia can also result from paraneoplastic processes. For example, T-cell lymphoma, classic Hodgkin lymphoma, SM, AML, MPN, and certain solid tumors (e.g., lung cancer, renal cell carcinoma, etc.) can abnormally release IL-2, IL-3, IL-5 or GM-CSF [91,92,93,94,95].

In addition to the clinical variants of HES already discussed, other myeloid neoplasms with associated HE must be excluded. AML with inv(16)(p13.1;q22) or t(16;16)(p13.1;q22) usually demonstrates eosinophils with characteristic coarse, baso-eosinophilic granules. MDS/MPN with neutrophilia (formerly atypical chronic myeloid leukemia, *BCR::ABL*-negative) can demonstrate eosinophilia in addition to leukocytosis, neutrophilic dysplasia, and frequent mutations involving *SETBP1*. CMML can demonstrate sustained monocytosis and myelodysplasia with or without elevated eosinophilia. For the latter, FISH for *PDGFRB* rearrangement should be evaluated to exclude MLN-TK. Of course, CEL should also be considered as a differential diagnosis if criteria for other myeloid neoplasms have not been met. Diagnostic criteria for CEL, specifically the time interval for sustained hypereosinophilia and evidence of both clonality and abnormal bone marrow morphology, have been updated in the fifth edition of the WHO Classification of Hematolymphoid neoplasms and International Consensus Classification (ICC) [53] (Table 3). 

## 7. Treatment

Managing HE depends on various factors including the underlying cause, clinical symptoms, and acuity of condition (Figure 3) [6,96]. The choice of treatment also varies depending on whether patients are asymptomatic or have organ damage. Glucocorticoid is typically the first line treatment for most forms of HE; however, management can differ in cases in which HE is caused by underlying parasitic disease, immunodeficiency syndrome or MLN-TK, such as MLN with *FIP1L1::PDGFRA* or steroid-resistant HE cases. For example, topical steroids with or without oral administration is the treatment of choice for eosinophilic dermatitis, while oral prednisone is used to manage localized eosinophilic infiltrates. Systemic cytoreduction therapy (hydroxyurea, methotrexate, and cyclophosphamide) and immunoadjuvants (e.g., interferon, and cyclosporin) are considered second-line therapeutic strategies for iHES and M-HES. Recent data showed that the majority patients (65–85%) with iHES, L-HES, and CEL were treated with first-line corticosteroids, whereas patients with M-HES were managed with TKI (e.g., imatinib) (81%); interferon alpha therapy was the optional regimen for those with CEL (42%) [71].

### 7.1. Conventional Therapy

The primary goal of therapy is to reduce the eosinophil count, which in turn alleviates inflammation and its related symptoms and prevents disease progression.

In some cases, watchful waiting may be appropriate, especially for familial eosinophilia or individuals with HE_US_ who remain symptom-free and show no clinical evidence of organ involvement. These patients should be monitored regularly for clinical progression.

In rare instances in which patients have extremely high eosinophil counts (>100 × 10^9^/L) or exhibit symptoms like leukostasis, heart failure, or thromboembolism, emergent management may be necessary. This may involve treatments like leukapheresis or high-dose steroids. High-dose steroids, such as prednisone given at 1 mg/kg or the equivalent, are a fundamental component of HES treatment, especially for patients without the *FIP1L1::PDGFRA* rearrangement. Steroids are believed to interfere with eosinophil production and lead to their sequestration. Before starting steroid therapy, it is important to exclude strongyloidiasis infection because as it could led to fatal parasitic dissemination with steroid therapy [97]. If clinical features suggest possible infestation, then administration of ivermectin (150 ug/kg orally daily × 1 dose) is warranted [4]. 

Steroid treatment is very effective and often leads to a response within 48 h and is considered as the mainstay of therapy; however, about one third of patients may still develop resistance to steroids. Some studies have been conducted to understand the mechanism of resistance to steroids. In 1989, one study established that abnormalities in the glucocorticoid receptor can lead to resistance [98]. Later, Stokes et al. established that very high levels of IL-5 may play a role in impairment of eosinophil apoptosis leading to resistance to glucocorticoid therapy [99]. For patients with steroid resistant disease, second line therapy using cytotoxic agents or immunomodulatory drugs should be considered. 

Imatinib, a TKI, is the first line of treatment for M-HES with *FIP1L1::PDGFRA* rearrangement and translocation involving *PDGFRB* [70]. These genetic alterations usually render patients resistant to steroids. However, imatinib has shown a rapid response and should be initiated with high-dose steroids in the first few days to prevent myocardial necrosis [13]. Treatment should not be delayed for molecular testing. High index of suspicion based on clinical features such as splenomegaly, high vitamin B_12_ levels or bone marrow showing dysplastic eosinophils and mast cells should prompt treatment initiation. The efficacy of second-generation TKI has been evaluated in small studies [100]. The role of imatinib in cases without these specific alterations is controversial with variable responses observed. Khoury et. al. performed a small prospective study that showed the response rate with imatinib was 54% in *PDGFRA*-negative M-HES and 0% among steroid refractory patients without M-HES [101]. For cases of severe eosinophilic granulomatosis with polyangiitis (EGPA), cyclophosphamide can be added to the induction therapy regimen along with a high-dose steroid [102].

### 7.2. Second-line Therapy

For second-line therapies, common options include hydroxyurea, interferon alpha, and cyclosporine. Hydroxyurea is often used alongside other treatments to reduce eosinophilic count for iHES or HES with overlapping rheumatological disorders, especially in steroid-resistant cases [13,103]. Hydroxyurea reduces cytokine production by Th_2_ cells and has a direct effect on eosinophils; however, its use is limited because of side effects. Cyclosporine inhibits T-cell activation and cytokine production. Cyclosporine is also effective for L-HES. Methotrexate serves as a steroid-sparing agent in EGPA and is better tolerated than cyclophosphamide because of fewer side effects [104]. Cyclophosphamide is administrated for patients with clinical presentation suggestive of eosinophilic granulomatosis with polyangiitis [4].

### 7.3. Disease-Specific Treatment

#### 7.3.1. iHES

Steroids are the first-line treatment of iHES with dosages adjusted according to the patient’s clinical state. For iHES patients who cannot tolerate high-dose steroids or do not respond well to first-fine therapy, additional medications should be considered (e.g., hydroxyurea, interferon-alpha or low-dose hydroxyurea). Other second-line choices include alemtuzumab, cyclosporine, or 2-chlorodeoxyadenosine; however, there is risk of cytotoxicity.

#### 7.3.2. Malignancy Associated HE

For patients who have HES associated with myeloid or lymphoid malignancies, treatment should focus on the underlying malignancy and select disease-specific treatment regimens (e.g., mepolizumab, reslizumab, and benralizumab). 

#### 7.3.3. Localized HE

If localized disease or a single organ is involved, only steroids or topical steroids should be considered. 

### 7.4. Novel Approaches and Clinical Trials

The studies of eosinophil-targeted regimen are limited (Table 5). Novel agents like mepolizumab and reslizumab (humanized monoclonal anti-IL-5 antibody) and benralizumab (IL-5 receptor antagonist) are newly studied biologics that demonstrated safety and efficacy for treating HE, particularly for those life-threatening situations or cases refractory to conventional therapy. Both mepolizumab and reslizumab lead to maturation arrest of eosinophils and significantly reduce the eosinophil count. Benralizumab is an IL-5 receptor alpha specific monoclonal antibody. It blocks the ligand IL-5 mediated independent pathway by high-affinity receptor binding; it induces apoptosis of eosinophils in blood, bone marrow and tissue via a direct antibody-dependent cell mediated cytotoxicity by binding to FcyRIII-alpha receptor of NK cells, and causing complete depletion of eosinophils, basophils, and their precursors. 

Mepolizumab was approved by the United States Food and Drug Administration (FDA) after a phase 3 trial for EGPA [105] (www.clinicaltrials.gov NCT00086658, accessed on 6 November 2023). It helped prolong remission and can be used as a steroid sparing agent. However, the benefits of mepolizumab in steroid-resistant cases and M-HES has not been seen [106]. Benralizumab was approved by the FDA for eosinophilic asthma and showed efficacy for eosinophilic GI disease [96]. A phase 3 study of its efficacy in HES is ongoing.

Dexpramipexole is another novel agent developed for treating amyotrophic lateral sclerosis, but has been found to cause eosinopenia [107]. Ongoing phase 2 studies showed promising results although the mechanism of dexpramipexole is uncertain [4,108]. AK002/lirentelimab is an afucosylated monoclonal antibody to SIGLEC-8 that causes maturational arrest of eosinophils [109]. Additionally, a monoclonal antibody targeting IL-4 receptor α, namely dupilumab is used for certain types of reactive HE such as asthma or active eosinophilic esophagitis (a phase 2 study) [110]. The roles of *JAK2* inhibitors (e.g., ruxolitinib) are currently under investigation in the context of M-HES (e.g., MLN with *PCM1::JAK2*) and L-HES treatment [111,112]. Stafiba, a *STAT5b* antagonist was developed to treat patients with patients with *STAT5b* mutation; omalizumab, a monoclonal anti-immunoglobulin E, is another option that targets IgE-mediated HE [66]. Similar to mepolizumab, alemtuzumab, an anti-CD52 antibody, has been adopted to treat HES-related symptoms with promising results [113]. The aforementioned key findings are summarized in Table 5.

### 7.5. Allogeneic Hematopoietic Stem Cell Transplant 

Stem cell transplant can be potentially curative for HES. Several case reports support the efficacy of allogenic stem cell transplant [114,115,116]. It is usually indicated for HES with multiorgan dysfunction or relapsed/refractory HES.

**Table 5 cancers-16-01383-t005:** Targeted Therapy for HE.

Pathway	Name	Nature/Function	Reference
IL-5 inhibitor	Mepolizumab and Reslizumab	Humanized monoclonal anti-IL-5 antibody	[105,106,117]
	Benralizumab	IL-5 receptor antagonist, focusing on eosinophils, basophils, and their precursors through antibody-dependent cell-mediated cytotoxicity	[117]
SIGLEC-8 inhibitor	Lirentelimab	Afucosylated antibody against Siglec-8 leading to decrease eosinophils and basophils and inactivate mast cells	[117]
IL-4 inhibitor	Dupilumab	IL-4 antibody, reduces IL-4 and IL-13 signaling, and blocks exotaxin-mediated tissue migration of eosinophils	[118,119]
TKI	Imatinib	*BCR-ABL1* inhibitor	[35,71]
	Ruxolitinib	*JAK2* inhibitor	[111]
*STAT5* inhibitor	Stafiba	Actively inhibiting the SH2 domains of *STAT5a* and *STAT5b*	[120]
IgE inhibitor	Omalizumab	Monoclonal anti-immunoglobulin E	[66]

Abbreviations: HE, hypereosinophilia; IL, interleukin; SH2, Src homology 2; SIGLEC, sialic-acid-binding immunoglobulin-like lectins; TKI, tyrosine kinase inhibitors.

## 8. Conclusions and Future Direction

HE encompasses a wide spectrum of differential diagnoses of reactive, paraneoplastic, hematological and non-hematological etiologies, proving a diagnostic challenge. The underlying mechanism vary among variants of HE, warranting advanced biological and molecular studies. Reactive HE should be always excluded before investigating other causes. A comprehensive clinical investigation and laboratory studies are needed before treatment options are selected. Be aware that secondary organ damage could occur in patients with HES, which require immediate assessment and therapy. TKI and/or targeted therapy should be considered in the treatment of MLN-TK as well as certain HE with specific mutations such as *STAT5b*.

## Figures and Tables

**Figure 1 cancers-16-01383-f001:**
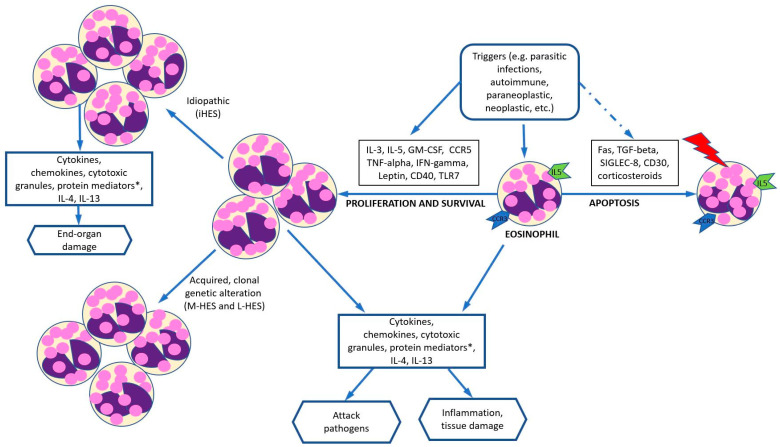
Proposed mechanisms of hypereosinophilia-related changes. With the appropriate trigger, cytokines or ligands (IL-3, IL-5, GM-CSF, tumor necrosis factor-alpha, interferon-gamma, leptin, CD40) can promote eosinophil proliferation and survival. Different cytokines or ligands (such as Fas, TGF beta, SIGLEC-8, CD30, and corticosteroids) can also facilitate eosinophil apoptosis. * Protein mediators include MBP, ECP, EPX, and eosinophil derived NGF. Abbreviations: CCR, chemokine receptor; CD, cluster of differentiation; ECP, eosinophil cationic protein; GM-CSF, granulocyte-macrophage colony-stimulating factor; HE, hypereosinophilia; HES, hypereosinophilic syndrome; EPX, eosinophil peroxidase; IG, immunoglobulin; IL, interleukin; L-HES, lymphocytic variant of hypereosinophilia; MBP, major basic protein; M-HES, myeloproliferative hypereosinophilia; NGF, nerve growth factor; SIGLEC-8: sialic acid-binding immunoglobulin-like lectin 8; TGF, transforming growth factor; TLR, toll-like receptor. Adapted from reference [14].

**Figure 2 cancers-16-01383-f002:**
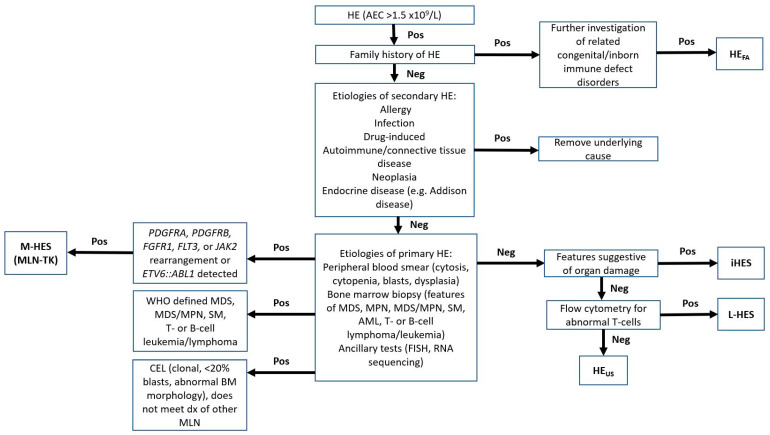
Diagnostic algorithm of hypereosinophilia. Abbreviations: AEC, absolute eosinophilic count, ALL, acute lymphoblastic leukemia; AML, acute myeloid leukemia; BM, bone marrow; CEL, chronic eosinophilic leukemia; dx, diagnosis; FISH, fluorescence in situ hybridization; HE, hypereosinophilia; HE_FA_, familial hypereosinophilia; HE_US_, hypereosinophilia of unknown significance; hx, history; iHES: idiopathic hypereosinophilia with organ damage; L-HES, lymphoid variant hypereosinophilia; MDS, myelodysplastic syndrome; M-HES, myeloid neoplasm-related hypereosinophilia; MLN, myeloid or lymphoid neoplasms; MLN-TK, myeloid/lymphoid neoplasm with eosinophilia and tyrosine kinase gene rearrangement; MPN, myeloproliferative neoplasm, Neg, negative; PB, peripheral blood; Pos, positive; SM, systemic mastocytosis. Adapted from reference [3].

**Figure 3 cancers-16-01383-f003:**
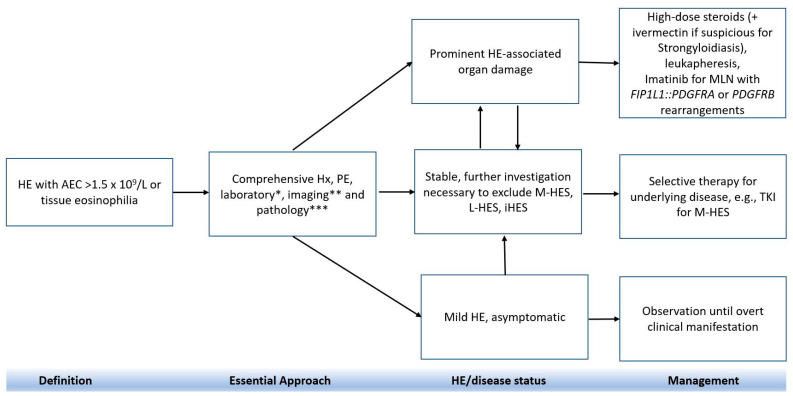
Stepwise treatment approach for patients with HE. Abbreviations: AEC, absolute eosinophil count; CBC, complete blood count; CMP, complete metabolic profile; FCM, flow cytometry analysis; FISH, fluorescent in situ hybridization; Hx, history; HE, hypereosinophilia; iHES, idiopathic hypereosinophilia; LFT, liver function test; L-HES, T-lymphocytic variant of hypereosinophilia; M-HES, myeloproliferative variants of hypereosinophilia; MLN, myeloid/lymphoid neoplasms with eosinophilia; TKI, tyrosine kinase inhibitor; PE, physical examination. * Laboratory tests including CBC, CMP, LFT, tryptase, vitamin B_12_, troponin, immunoglobulin, flow cytometry, etc. ** Instrumental or imaging studies including electrocardiogram, echocardiogram, computed tomography (CT) or positron emission tomography (PET) scan, magnetic resonance imaging (MRI), pulmonary function tests, etc. *** Pathology including peripheral blood smear, bone marrow and/or tissue biopsy and associated cytogenetics, FISH, and molecular studies. Adpted from [13].

**Table 1 cancers-16-01383-t001:** Common subcategories of HE.

Primary/Clonal HE		Secondary/Reactive HE	
Category	Example	Category	Example
Myeloproliferative variants of HE (M-HES)	AML ALL MPN (CEL) MLN-TK (*PDGFRA*, *PDGFRB*, *FGFR1*, *JAK2*, *FLT3* rearrangement, and *ETV6::ABL1* fusion and other variants)	Fungal, parasite, protozoal, viral, or mycobacterial infection	Systemic fungal infection (*Coccidioides*), helminth infection (tissue invasion phase), ectoparasites, *Sarcocystis* and *Cystoisospora*, HIV, COVID-19, tuberculosis (rare)
T-lymphocytic variant (L-HES)	L-HES	Atopic/Allergy	Atopic dermatitis, chronic rhinosinusitis, asthma, drug hypersensitivity
** Idiopathic HE **		Autoimmune and immune dysregulation	Connective tissue disorders, UC, Crohn’s disease, IgG4-related disease, sarcoidosis
**Category**	**Example**	Neoplasm	Leukemia, lymphoma (cHL, T-cell), solid tumor
Idiopathic HES (iHES)	HE with organ damage	Therapy/medication	Radiation, interleukin or GM-CSF therapy
HE of unknown significance (HE_US_)	HE without organ damage and unknown etiology	Other	Addison’s disease, cholesterol emboli
Organ-restricted HE	Eosinophilic esophagitis, Eosinophilic gastrointestinal disorders, Eosinophilic dermatitis, Chronic eosinophilic pneumonia	**Family HE/Inborn errors** **in immunity (IEIs) ***	Omenn syndrome, Wiskott-Aldrich syndrome, Netherton syndrome, and Hyper IgE syndrome (*DOCK8* deficiency), Loeys-Dietz syndrome

Abbreviations: ALL, acute lymphoblastic leukemia; AML, acute myeloid leukemia; ALL, acute lymphoblastic leukemia; COVID, coronavirus disease; CEL, chronic eosinophilic leukemia; cHL, classic Hodgkin lymphoma; GM-CSF, granulocyte-macrophage colony-stimulating factor; HE, hypereosinophilia; HES; hypereosinophilia syndrome; HE_US_, HE of unknown significance; HIV, human immunodeficiency virus; Ig, immunoglobulin; IL, interleukin; L-HES, lymphoid-variant hypereosinophilia; M-HES, myeloproliferative variants of HE; MLN-TK, myeloid/lymphoid neoplasms with eosinophilia and tyrosine kinase gene fusion; MPN, myeloproliferative neoplasm; UC, ulcerative colitis. * These patients (IEIs) may have mild eosinophilia but tend to have a benign course and are not associated with a neoplastic process. Some categorize them as a familial HE.

**Table 2 cancers-16-01383-t002:** Diagnosis of hypereosinophilia-induced organ damage and dysfunction [15].

	Symptoms and Signs	Functional Tests	Tissue Biopsy
Cardiac (myocarditis)	Dyspnea, arrythmia, ischemic attack	Serum troponin ECG, EchoCG, MRI	N/A
Lung	Dyspnea, hypoxemia, eosinophilic pleural effusions	CXR, chest CT, pulmonary function testing	Bronchoalveolar lavage, Lung bx, Pleural fluid cytology
GI tract (eosinophilic esophagitis, gastritis, enteritis, colitis)	Esophagus: reflux, dysphagia Bowel: abdominal pain, GI bleed, ischemia	Serum LFT, amylase, lipase	Endoscopic tissue bx
Cutaneous	Urticaria, angioedema, rash, erythematous papules, or nodules	N/A	Skin bx
Renal	Chronic UTI-like symptoms	Serum creatinine level, Urine eosinophils	Kidney bx
Neurologic	Peripheral neuropathy, TIA, stroke	Head MRI, Head CT, Nerve conduction studies	Nerve biopsy (rarely performed)

Abbreviations: bx, biopsy; CT, computed tomography; CXR, chest radiograph; GI, gastrointestinal; ECG, electrocardiogram; EchoCG, echocardiogram; GI, gastrointestinal; LFT, liver function test; MRI, magnetic resonance imaging; N/A, not applicable; TIA, transient ischemic attack; UTI, urinary tract infection.

**Table 3 cancers-16-01383-t003:** Critical features and diagnostic clues of reactive and neoplastic eosinophilia [25,26].

	Clinical	Laboratory/ Pathology	Molecular/ Genetic Study
**Secondary (reactive)**			
Allergy	Including atopic or non-atopic diseases: ABPA, asthma, allergic rhinitis, ECRS, NARES, food allergies, atopic dermatitis, drug allergies (e.g., DRESS), eosinophilic otitis media, eosinophilic laryngitis	Elevated IgE level	N/A
Infection	Infected microorganism related signs and symptoms Parasitic (*Toxocara*, *Toxoplasma*, *Strongyloides*, *Ascariasis*, *Trichinella*, *Echinococcus*, scabies, microfilariae), Fungal (*Coccidioides*), Viral (HIV, HCV)	Positive culture of microorganisms, elevated viral load or antibody titers, identification of parasites	PCR or NGS positive for specific microorganisms
Autoimmune	Connective tissue disorders, sarcoidosis, IBD, bullous pemphigoid, systemic vasculitis, granulomatosis with polyangiitis, eosinophilic granulomatosis with polyangiitis (Churg–Strauss syndrome)	Depending on disease type, presence of rheumatoid factor, ANA, anti-dsDNA, etc.	N/A
Immunodeficiency	Hyper IgE syndrome (Job syndrome), Omenn syndrome	Markedly elevated IgE level	*STAT3* mutations (Job syndrome); *RAG1* and *RAG2* mutations (Omenn syndrome)
Organ specific HE	Esophagitis (dyspepsia, dysphasia, reflux), gastroenteritis, cystitis, pneumonia (cough), dermatologic conditions (rash, pruritis)	Tissue infiltration by eosinophils, infectious or neoplastic etiologies have been excluded	N/A
Therapy/ Medication	Radiation, IL-2, IL-3, IL-5, or GM-CSF	N/A	N/A
Endocrine disorders	Addison’s disease	Decreased aldosterone, increased ACTH	N/A
Rare diseases	Gleich syndrome (episodic angioedema, eosinophilia, polyclonal IgM) Eosinophilia-myalgia syndrome	Eosinophilia, polyclonal IgM for Gleich syndrome	N/A
Other	GvHD, cholesterol embolization, radiation exposure	GvHD specific findings	Post-engraftment analysis for GvHD
**Primary (clonal)**			
MLN-TK	Variable, male predominance, hepatosplenomegaly, anemia; Good response to imatinib or other TKI, Variable steroid response	Concurrent or subsequent myeloid and lymphoid neoplasms; increased serum B_12_, thrombocytopenia, dysplastic eosinophils ± myelofibrosis, leukoerythroblastosis	FISH, RT-PCR or RNA sequencing for *PDGFRA* (*CHIC* deletion), *PDGFRB*, *FGFR1*, *JAK2*, or *FLT3* fusions, and *ETV6::ABL1* fusion
CEL	Asymptomatic or symptomatic (B-symptoms), systemic involvement	Eosinophilia > 1.5 × 10^9^/L on at least two occasions over an interval of 4+ weeks, clonality identified, abnormal BM morphology, <20% blasts. Excludes: CHIP, MPN, MDS/MPN, MDS, MLN-TK, SM and AML with inv(16). Tissue eosinophilic infiltrate can be seen	Clonal abnormalities, e.g., mutations involving *ASXl1*, *DNMT3A*, *EZH2*, *TET2*, *SRSF2*, *SETBP1*, and *CBL* (VAF ≥ 10%, more than one mutation preferred)
*KIT*-mutated SM	Depends on affected organ or tissue, asymptomatic to pruritic (skin), diarrhea (GI), organomegaly	Increased serum tryptase	PCR or NGS for *c-KIT* mutation
Lymphoid variant of HE (L-HES)	Male = Female, may manifest with skin lesions, GI symptoms, or obstructive lung disease; potential progression to T-cell lymphoma, rare cardiac involvement, responds to steroids with good outcome	Abnormal T-cell population (often sCD3^−^/CD4+), increased IL-4 and IL-5 levels, increased serum IgE, increased TARC (thymus activation regulated chemokine)	Clonal TCR gene rearrangement detected by PCR
**Paraneoplastic HE**			
AML	Cytopenia related signs and symptoms (e.g., pallor, infection, bleeding)	Circulating blasts ± HE; bone marrow with increased blasts (can be < 20%), immature eosinophilic precursors	inv(16), t(16;16)/*CBFB::MYH11*
B-ALL	Cytopenia related signs and symptoms (e.g., pallor, infection, bleeding), ±lymphadenopathy or splenomegaly	PB and BM loaded with B-lymphoblasts and increased eosinophils	t(5;14)(q31.1;q32.3) /*IGH::IL3*
MDS	Asymptomatic, weakness, fatigue	Cytopenia, thrombocytosis seen in MDS with del(5q); bone marrow dysplasia ≥ 10% of each lineage, ±increased blasts	Del(5q)/-5, del(7q)/-7, +8, del(20q)/-20, del(17p)/-17, *KMT2A/MLL* rearrangement *SF3B1* or *TP53* mutation *
MDS/MPN	±Splenomegaly	Leukocytosis, commonly monocytosis or thrombocytosis, BM with mixed myelodysplastic and myeloproliferative features	Cytogenetic alterations related to MDS and/or MPN, molecular changes related to *SF3B1*, *JAK2*, *CALR*, *MPL* can be identified
SM, MPN other than CEL	Pruritis for SM, splenomegaly	Leukocytosis for CML; elevated serum tryptase level for SM	*BCR::ABL1* for CML, *c-KIT D816V* or other variants for SM, *JAK2*, *CALR*, or *MPL* mutation
cHL	B-symptoms, lymphadenopathy	Tissue biopsy with Reed-Sternberg or Hodgkin cells positive for CD30, CD15, dim PAX-5 and MUM1 and negative for CD20, and CD45	B-cell gene rearrangement by PCR
LCH	Lytic bone lesions, skin lesions	Tissue biopsy with Langerhans cell proliferation and eosinophilic infiltrate, positive for CD1a, S100, Langerin, +/− BRAF	40% with *BRAF* V600E
T-cell neoplasms	AILT, PTCL	Sheets of abnormal proliferation of neoplastic T-cells, along with reactive histiocytes, or plasma cells and EBV+ B-cells (AILT)	*TET2*, *ROA* mutation
Non-hematologic malignancies	Adenocarcinoma of the lung, gastrointestinal tract, pancreas, thyroid, genital and skin tumors	Solid tumor confirmed by tissue biopsy, elevated cancer markers (e.g., CEA, CA19.9, TSH)	Genomic study for gene alteration specific to the tumor
**Idiopathic**			
iHES	Variable, mild to intensive pruritus, angioedema, accompany with organ damage related signs or symptoms	HE or tissue eosinophilic infiltrate, does not fulfill the dx criteria of reactive or neoplastic HE + eosinophilic organ damage	No clonal abnormality
HE_US_	Nonspecific or asymptomatic; no evidence of eosinophilic organ damage-associated signs and symptoms	HE or tissue eosinophilic infiltrate, not fulfilling the dx criteria of reactive or neoplastic HE	None

* *TP53* mutation: interpretation upon ICC or the fifth edition WHO classification. Abbreviations: ABPA, allergic bronchopulmonary aspergillosis; ACTH: adrenocorticotrophic hormone; AILT, angioimmunoblastic T-cell lymphoma; AML: acute myeloid leukemia; ANA: anti-nuclear antibody; B-ALL, acute B-lymphoblastic leukemia/lymphoma; BM, bone marrow; CEA, carcinoembryonic antigen; CEL: chronic eosinophilic leukemia; CHIP, clonal hematopoiesis of indeterminate potential; cHL, classic Hodgkin lymphoma; CML, chronic myeloid leukemia; DRESS, drug reaction with eosinophilia and systemic symptoms; dsDNA, double stranded deoxyribonucleic acid; dx, diagnostic; EBV, Epstein-Barr virus; ECRS, eosinophilic chronic rhinosinusitis; GM-CSF, granulocyte-macrophage colony-stimulating factor; GvHD: graft-versus-host disease; HCV, hepatitis C virus; HE, hypereosinophilia; HEus: hypereosinophilia of unknown significance; HIV, human immunodeficiency virus; IBD, inflammatory bowel disease; ICC, intrahepatic cholangiocarcinoma; Ig, immunoglobulin; iHES: idiopathic hypereosinophilic syndromes; IL, interleukin; LCH, Langerhans cell histiocytosis; L-HES, lymphoid variant of HE; MDS, myelodysplastic neoplasm; MDS/MPN, myelodysplastic/myeloproliferative neoplasms; MLN-TK, myeloid/lymphoid neoplasm with eosinophilia and tyrosine kinase gene rearrangement; MPN, myeloproliferative neoplasm; N/A, not applicable; NARES, nonallergic rhinitis with eosinophilia syndrome; NGS, Next-generation sequencing; PB, peripheral blood; PCR, polymerase chain reaction, PTCL, peripheral T-cell lymphoma; SM, systemic mastocytosis; TARC, thymus activation-regulated chemokine; TCR, T-cell receptor; TSH, thyroid-stimulating hormone; VAF, variant allele frequency.

## Data Availability

Not applicable.

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
