# Peer review of "Clinical and Therapeutic Intervention of Hypereosinophilia in the Era of Molecular Diagnosis"

_cancers, 2024, doi:10.3390/cancers16071383_

Round 1

Reviewer 1 Report

Comments and Suggestions for Authors

The authors present a thorough and accurate review of eosinophilia and hypereosinophilic syndrome

Additional information would make it even better: discussion of age and eosinophilia, expand on parasitic diseases and exposures beyond Strongyloides,  expand on drug-induced eosinophilia --as this is more common, discuss eosinophilopoiesis  and wouldn't the molecular lesions be limited to. arising in eosinophil progenitors?

Line 393: I am not aware of Kostmann (which is an old term -- specifically it is HAX1 mutation) or severe congenital neutropenia be associated with eosinophilia.

More discussion of CEL.  

Comments on the Quality of English Language

A few syntax errors

1. line 28 deep insights, not deep sights

2. lines 178-179, the clinical history of secondary eosinophilia should always be excluded at first --- what is meaning secondary eosinophilia should be excluded by the clinical history... 

3. Figure 2 in text: AEC not ACE, and abbreviation must be defined in legejnd

4. line 280 FIP1L1::PDGFRA not the other way around

5. line 305  - why is the sentence italicized? for emphasis - then I would break it out into a new paragraph

6. lines 358-359 needs a preposition to start the sentence. or break into two sentences

7. line 483: one-third not 1 third

Author Response

Thank you for dedicating your time to assess the manuscript and sharing insightful suggestions. Enclosed are our detailed responses, along with the highlighted revisions in the resubmitted document.

Reviewer 1

Comment 1: 

Additional information would make it even better: discussion of age and eosinophilia, expand on parasitic diseases and exposures beyond Strongyloides, expand on drug-induced eosinophilia --as this is more common, discuss eosinophilopoiesis  and wouldn't the molecular lesions be limited to. arising in eosinophil progenitors?

Response: Thank you for the suggestions. We acknowledge the significance of discussing age and eosinophilia; however, it's important to note that demographic factors such as age may vary depending on the underlying cause of eosinophilia.

We have tried to summarize the most commonly seen parasites associated with eosinophilia in Table 1 and Table 3. 

Comment added on drug induced eosinophilia on page 13, paragraph 6.8, line 457-460. "Administration of cytokines IL-2, IL-3, IL-5, or GM-CSF as medication can also elevate the number of eosinophils in blood. Additionally, any medication can cause a drug hypersensitivity reaction or drug-associated eosinophilia."

Sorry for not understanding the comment properly. I am uncertain if I grasp which particular abnormalities you're referring to.

Comment 2: 

Line 393: I am not aware of Kostmann (which is an old term -- specifically it is HAX1 mutation) or severe congenital neutropenia be associated with eosinophilia.

Response: We have changed Kostmann disease to severe congenital neutropenia on page 13, paragraph 6.6, line 435. "The defects can be caused by (1) antibody deficiencies (e.g., common variable immunodeficiency disorder), or combined immunodeficiencies (e.g., Omenn syndrome, Wiskott-Aldrich syndrome, Netherton syndrome, or hyper IgE syndrome); (2) diseases associated with dysregulation of cellular immunity (autoimmune lymphoproliferative syndrome [ALPS]);  or (3) dysregulation of phagocytosis (severe congenital neutropenia, formerly Kostmann disease)".

Comment 3. 

a. line 28 deep insights, not deep sights : Changed on page 1, paragraph 2, line 28

b. lines 178-179, the clinical history of secondary eosinophilia should always be excluded at first --- what is meaning secondary eosinophilia should be excluded by the clinical history: Have changed to " Secondary cause of eosinophilia should always be excluded first. The clinical history must be carefully reviewed for allergies, current medications" on page 5, paragraph 5.1, line 178.

c. Figure 2 in text: AEC not ACE, and abbreviation must be defined in legend: Correction made in figure 2 on page 5.

d.line 280 FIP1L1::PDGFRA not the other way around: Correction has been made. Currently on page 10 paragraph 3 and line 304

e. line 305 - why is the sentence italicized? for emphasis - then I would break it out into a new paragraph: We have broken these into a new paragraph and un-italicized on page 10, paragraph 6, line 329.

f. lines 358-359 needs a preposition to start the sentence. or break into two sentences. We have changed to "Not much is known about the molecular mechanism. T-helper type 2 (Th2) cytokines (IL-4, IL-13 and GM-CSF) may be involved in the increased serum IgE production and polyclonal hypergammaglobulinemia" on page 12, paragraph 6.2, line 381-383.

g. line 483: one-third not 1 third: Changed on page 15, paragraph 7.1, line 522. 

Reviewer 2. 

Comment 1: 1.     Figure 1. I know it is a figure, but one could make the colors of the nucleus and especially the granules in eosinophils more realistic. 

Response: The color was changed to pink and violet to make it more realistic in Figure 1 on page 4. 

Comment 2: Some recent literature on the role of eosinophils in allergies, especially their most “eosinophilic” forms, could be incorporated in the respective place (e.g. PMID: 34947981, 36835081, 33260893).

Response: We have added a paragraph on potential genetic determinants and biomarkers on page 7, paragraph 5.3. We have included the references as well. 

"5.3. Potential genetic determinants and biomarkers:

Extracellular vesicles and a number of potential blood-based biomarkers are being investigated to aid in diagnosis and treatment of atopic conditions [16].  Eosinophils have been noted to carry microRNA (miRNA) in extracellular vesicles to other cells, potentially playing a role in gene regulation and expression [16]. One study sequenced the eosinophil transciptome in individuals with atopic conditions and compared them to healthy controls. They discovered at least 18 miRNAs were differentially expressed in individuals with allergic conditions when compared to those unaffected [17]. Additionally, genome-wide association studies (GWAS) have identified multiple genes (chromosomes 1q23 [FCER1A], 5q31 [RAD50, IL13, IL4], and 12q13 [STAT6] and loci 6p21.3 [HLA-DRB1] and 16p12 [IL4R, IL21R]) that may influence the regulation of IgE and serum IgE levels [18].  Interestingly, the relationships of total serum IgE levels and atopic conditions may not be as simple as initially thought.  Specific atopic conditions (asthma, allergies rhinitis, atopic dermatitis) show little overlap with genetic determinants of total serum IgE levels, suggesting that elevated serum IgE levels may be an epiphenomenon [19-21],.  A stronger association may be seen in specific IgE with allergic conditions rather than total serum IgE levels [22-24]. Studies are ongoing, but potential genetic markers and biomarkers are on the horizon."

Comment 3: Recently, one speaks rather on “type 2” than “Th2” response or immunity.

Response: Thank you for your recommendation. Our corresponding author is currently abroad and unavailable to provide input on this matter. However, we don't have a strong preference either way. Therefore, we have opted to retain "Th2 immune response" for the time being.

Comment 4: .     While writing on HIES, IgE and allergies, one should make it clear that the genetic susceptibility loci/genetic determinants of the three traits substantially differ (PMID: 22909159).

Response: Please see the response of comment 2. We were uncertain about the appropriate section to integrate the concepts from suggestions 2 and 4. Therefore, we combined them under a new heading. 

Reviewer 2 Report

Comments and Suggestions for Authors

With interest, I read the manuscript cancers-2895193. In my view, it is a nice work. Thus, I have only minor and/or facultative comments/suggestions only.

1.     Figure 1. I know it is a figure, but one could make the colors of the nucleus and especially the granules in eosinophils more realistic.

2.     Some recent literature on the role of eosinophils in allergies, especially their most “eosinophilic” forms, could be incorporated in the respective place (e.g. PMID: 34947981, 36835081, 33260893).

3.     Recently, one speaks rather on “type 2” than “Th2” response or immunity.

4.     While writing on HIES, IgE and allergies, one should make it clear that the genetic susceptibility loci/genetic determinants of the three traits substantially differ (PMID: 22909159).

Author Response

Reviewer 2. 

Comment 1:  Figure 1. I know it is a figure, but one could make the colors of the nucleus and especially the granules in eosinophils more realistic. 

Response: The color was changed to pink and violet to make it more realistic in Figure 1 on page 4. 

Comment 2: Some recent literature on the role of eosinophils in allergies, especially their most “eosinophilic” forms, could be incorporated in the respective place (e.g. PMID: 34947981, 36835081, 33260893).

Response: We have added a paragraph on potential genetic determinants and biomarkers on page 7, paragraph 5.3. We have included the references as well. 

"5.3. Potential genetic determinants and biomarkers:

Extracellular vesicles and a number of potential blood-based biomarkers are being investigated to aid in diagnosis and treatment of atopic conditions [16].  Eosinophils have been noted to carry microRNA (miRNA) in extracellular vesicles to other cells, potentially playing a role in gene regulation and expression [16]. One study sequenced the eosinophil transciptome in individuals with atopic conditions and compared them to healthy controls. They discovered at least 18 miRNAs were differentially expressed in individuals with allergic conditions when compared to those unaffected [17]. Additionally, genome-wide association studies (GWAS) have identified multiple genes (chromosomes 1q23 [FCER1A], 5q31 [RAD50, IL13, IL4], and 12q13 [STAT6] and loci 6p21.3 [HLA-DRB1] and 16p12 [IL4R, IL21R]) that may influence the regulation of IgE and serum IgE levels [18].  Interestingly, the relationships of total serum IgE levels and atopic conditions may not be as simple as initially thought.  Specific atopic conditions (asthma, allergies rhinitis, atopic dermatitis) show little overlap with genetic determinants of total serum IgE levels, suggesting that elevated serum IgE levels may be an epiphenomenon [19-21],.  A stronger association may be seen in specific IgE with allergic conditions rather than total serum IgE levels [22-24]. Studies are ongoing, but potential genetic markers and biomarkers are on the horizon."

Comment 3: Recently, one speaks rather on “type 2” than “Th2” response or immunity.

Response: Thank you for your recommendation. Our corresponding author is currently abroad and unavailable to provide input on this matter. However, we don't have a strong preference either way. Therefore, we have opted to retain "Th2 immune response" for the time being.

Comment 4: .     While writing on HIES, IgE and allergies, one should make it clear that the genetic susceptibility loci/genetic determinants of the three traits substantially differ (PMID: 22909159).

Response: Please see the response to comment 2. We were uncertain about the appropriate section to integrate the concepts from suggestions 2 and 4. Therefore, we combined them under a new heading.